# Gender Effects on Left Ventricular Responses and Survival in Patients with Severe Aortic Regurgitation: Results from a Cohort of 756 Patients with up to 22 Years of Follow-Up

**DOI:** 10.3390/medsci11020036

**Published:** 2023-05-23

**Authors:** Padmini Varadarajan, Ramdas G. Pai

**Affiliations:** 1Division of Cardiology, Loma Linda University Medical Center, Loma Linda, CA 92521, USA; 2Division of Cardiology, Riverside School of Medicine, University of California, Riverside, CA 92521, USA

**Keywords:** gender, aortic regurgitation, aortic valve replacement, survival

## Abstract

Objectives: We sought to evaluate the effect of gender on biology, therapeutic decisions, and survival in patients with severe aortic regurgitation (AR). Background: Gender affects adaptive response to the presence of valvular heart diseases and therapeutic decisions. The impact of these on survival in severe AR patients is not known. Methods: This observational study was compiled from our echocardiographic database which was screened (1993–2007) for patients with severe AR. Detailed chart reviews were performed. Mortality data were obtained from the Social Security Death Index and analyzed as a function of gender. Results: Of the 756 patients with severe AR, 308 (41%) were women. Over a follow-up of up to 22 years, there were 434 deaths. Women compared to men were older (64 ± 18 vs. 59 ± 17 years, *p* = 0.0002). Women also had smaller left ventricular (LV) end diastolic dimension (5.2 ± 1.1 vs. 6.0 ± 1.0 cm, *p* < 0.0001), higher EF (56% ± 17% vs. 52% ± 18%, *p* = 0.003), higher prevalence of diabetes mellitus (18% vs. 11%, *p* = 0.006), and higher prevalence of ≥2+ mitral regurgitation (52% vs. 40%, *p* = 0.0008) despite a smaller LV size. Women were also less likely to receive aortic valve replacement (AVR) (24% vs. 48%, *p* < 0.0001) compared to men and had a lower survival on univariate analysis (*p* = 0.001). However, after adjusting for group differences including AVR rates, gender was not an independent predictor of survival. However, the survival benefit associated with AVR was similar in both women and men. Conclusions: This study strongly suggests that female gender is associated with different biological responses to AR compared to men. There is also a lower AVR rate in women, but women derive similar survival benefit as men with AVR. Gender does not seem to affect survival in an independent fashion in patients with severe AR after adjusting for group differences and AVR rates.

## 1. Background

Valvular heart disease is present in 2.5% of the United States population with no gender difference. Aortic regurgitation (AR) is seen in about 0.5% of the United States population [1]. Valvular lesions cause either a pressure or volume overload of the heart leading to cardiac hypertrophy or dilation. In pressure overload conditions such as aortic stenosis, it has been demonstrated that women develop more concentric left ventricular (LV) hypertrophy and have preserved ejection fraction with less severe myocardial fibrosis [2,3]. Volume overload conditions such as aortic regurgitation (AR) induce eccentric hypertrophy. There is lack of data in humans evaluating the effect of gender on volume overload conditions such as AR and associated outcomes. Hence, we sought to evaluate the effect of gender on survival and remodeling responses in patients with severe AR with an extended follow-up of up to 22 years.

## 2. Methods

### 2.1. Study Population

This was a retrospective observational study conducted in a large university medical center. The study was approved by the institutional review board, waiving the need for patient consent. The echocardiographic database was searched for patients with severe AR from 1993 to 2007 yielding a total of 786 unique patients, 30 of whom had no follow-up and were excluded from the study. The final study cohort comprised 756 patients. Extended follow-up of up to 22 years was obtained. Severe AR was diagnosed by a Level 3-trained echocardiographer as per published guidelines [4]. Comprehensive chart review was performed on these patients. Some of the short- and medium-term results from this cohort have been published previously but not with respect to gender [5,6,7].

### 2.2. Clinical Variables

Clinical comorbidities consisting of hypertension, diabetes mellitus, renal insufficiency, and coronary artery disease (CAD) were recorded as reported in our previous studies [5,6,7].

### 2.3. Echocardiography

The LV ejection fraction (EF) was assessed by planimetry or visually by a Level 3-trained echocardiographer from a standard 2-dimensional echocardiogram. This has been shown to be reliable and has been validated against contrast and radionuclide LV angiography [8,9]. Recommendations of the American Society of Echocardiography were followed when computing anatomic measurements [10].

### 2.4. Pharmacological Data

Pharmacotherapy around the time of initial echo was recorded and placed into broad categories of various cardiac medications as described previously [5,6,7]. Patients were considered to be on a pharmacological agent only if they received it for at least 1 month’s duration.

### 2.5. Mortality Data

All-cause mortality was the end point of the study. Mortality data were obtained from the National Death Index using social security numbers.

### 2.6. Statistical Methods

Stat View version 5.01 (SAS Institute, Cary, NC, USA) program was used for statistical analysis. Kaplan–Meier survival curves were computed for patients with severe AR based on gender using the log-rank statistic. Characteristics of male and female patients were compared using the Student t-test for continuous variables and the chi-squared test for categorical variables. Cox proportional hazards models were used to adjust for comorbidities and covariate imbalances. A *p* value of 0.05 was considered significant.

## 3. Results

### 3.1. Baseline Characteristics of the Whole Cohort

A total of 756 patients had severe AR. The baseline characteristics of these patients were mean age 61 ± 18 years, 59% men, mean LVEF 54 ± 19%, and diabetes mellitus in 14%, hypertension in 65%, and CAD in 33%. Over a period of 22 years, there were 434 deaths. The likely causes of AR based on echocardiographic appearance and chart review were as follows: bicuspid aortic valve in 78 (10%), dilated aortic root in 79 (10%), degenerative or calcific aortic valve disease in 220 (30%), and prior infective endocarditis in 78 (10%) patients. The rest had mixed mechanisms.

### 3.2. Baseline Characteristics as a Function of Gender and Gender Differences in the Remodeling Response

In total, 308 (41%) of the 756 patients with severe AR were women. The women compared to the men were older (64 ± 18 vs. 59 ± 17 years, *p* = 0.0002), had smaller LV end diastolic dimension (5.2 ± 1.1 vs. 6.0 ± 1 cm, *p* < 0.0001), had lesser degree of LV hypertrophy (LV posterior wall thickness 1.1 ± 0.2 vs. 1.2 ± 0.2 cm, *p* = 0.002, interventricular septum 1.2 ± 0.3 vs. 1.3 ± 0.2, *p* 0.0008, respectively), had higher LV ejection fraction (56 ± 17 vs. 52 ± 18%, *p* = 0.003), had higher prevalence of diabetes mellitus (18% vs. 11%, *p* = 0.006), and had lower rates of aortic valve replacement (AVR) (24% vs. 48%, *p* < 0.0001) when compared to males (Table 1). Despite having smaller ventricles with preserved EF, women had higher prevalence of ≥2+ MR (52% vs. 40%, *p* = 0.0008) when compared to men (Table 1).

### 3.3. Gender and Survival

Over a mean follow-up of 7.9 years, female gender was associated with a lower survival rate (5-, 10-, and 20-year survival rates of 59%, 42%, and 30%, respectively) compared to males (5-, 10-, and 20-year survival rates of 63%, 52%, and 50%, respectively) (*p* = 0.001) (Figure 1). However, after adjusting for group differences and AVR rates (24% in women and 48% in men) in Cox regression analysis, gender was not an independent predictor of survival (Table 2).

### 3.4. AVR Rates and Survival as a Function of Gender

Of the 308 female patients, only 74 (24%) had AVR. Patients with AVR had better 5-, 10-, and 20-year survival (80%, 65%, and 53% vs. 50%, 38%, and 28%, *p* < 0.0001) compared to those who did not undergo AVR (Figure 2)**.** Men who underwent AVR also had better 5-, 10-, and 20-year survival (79%, 62%, and 58% vs. 52%, 40%, and 38%, *p* < 0.0001) compared to those who did not undergo AVR (Figure 3).

### 3.5. Gender Effects on Survival in Nonsurgical Patients

In the non-surgical group 5-, 10-, and 20-year survival was 58%, 45%, and 38% in men compared to 52%, 40%, and 28%, respectively, in women (*p* = 0.04). However, the women were older (age 67.1 ± 16.4 vs. 63.2 ± 16.8 year, *p* = 0.01), had smaller LVEDD (5.1 ± 1.0 vs. 5.9 ± 0.9 cm, *p* <0.0001), smaller LVESD (3.5 ± 1 vs. 4.2 ± 0.07, *p* < 0.0001), lower septal wall thickness (1.1 ± 0.2 vs. 1.2 ± 0.2, *p* = 0.03), lower posterior wall thickness (1.1 ± 0.2 vs. 1.2 ± 0.2, *p* = 0.05), and higher ejection fraction 54 ± 18% vs. 51 ± 19%, *p* = 0.03) when compared to men. After adjusting for these group differences, gender was not an independent predictor of survival in the non-surgically managed patients.

### 3.6. Gender Effects on Survival in Surgical Patients

The 5-, 10-, and 20-year survival in women with AVR was 80%, 65%, and 53%, respectively, compared to 79%, 62%, and 58% in men who had AVR (Figure 4). The survival in women vs. men was similar in both univariate and multivariate analysis.

## 4. Discussion

Our study shows that the women with severe AR are older, have differences in LV adaptive responses, and have lower AVR rates but have similar survival rates after adjusting for these group differences.

### 4.1. Gender and Aortic Valve Disease

Gender differences in valve lesions such as aortic stenosis, especially with respect to remodeling characteristics, have been extensively studied in the literature. It is well known that aortic stenosis results in small, hypertrophied left ventricle with preserved ejection fraction in females which has implications for surgical management. However, there is a paucity of data with regards to survival, remodeling characteristics, and rates of aortic valve replacement in females with aortic regurgitation.

### 4.2. Gender and Volume Overload States

While the biological effects of AR have not been well studied in humans, there are few animal studies evaluating the effect of volume overload. Dent et al. studied the gender differences in cardiac dysfunction and remodeling in a rat model of LV volume overload produced by AV fistula [11]. They found that female rats had greater LV hypertrophy and better EF compared to males. Ovariectomy resulted in a decrease in LV contractility which was restored by estrogen therapy. In our study, women had higher EF as well. Beaumont et al. studied the differences in myocardial transcriptional adaptations to AR with respect to gender differences in a rat model [12]. These investigators found that female rats showed relatively higher LV mass increase compared to males but similar increases in LV size. The fatty acid oxidation (FAO)-related LV enzyme activity and mitochondrial biogenesis were reduced in male rats but well preserved in females, supporting better LV function and survival in the female rats. Women in our study had smaller left ventricles and higher ejection fractions consistent with these animal models. However, our patients were older and mostly post-menopausal without the confounding effects of estrogen or ovarian function.

### 4.3. Aortic Valve Replacement in Females

Studies have shown that AVR utilization in females is consistently lower than in males. In a study by Chaker et al. consisting of 16,809 patients with aortic valve disease, 63% of males had AVR compared to 37% of females. They also reported higher in-hospital mortality for females undergoing AVR [13]. Hamed et al. reported in their study consisting of 406 patients, of which 183 were females, that gender had no impact on survival post AVR after correcting for confounding variables [14], which is a finding similar to ours. This finding is similar to our study confirming lower AVR rates in women compared to men.

## 5. Summary and Conclusions

Our study shows that females with severe AR were older, had differential remodeling characteristics such as smaller chamber sizes, had less thick ventricles, had better LVEF, and had lower rates of AVR compared to males. Female subjects had worse survival when compared to males on univariate analysis, which did not translate into statistical significance on multivariate analysis. Our study also shows that females tend to have lower rates of AVR which is similar to the study by Chaker et al. However, AVR seems to offer a survival benefit in women similar to men. Based on our study, females have lower rates of AVR with survival being similar to males. Hence, females with severe AR should be offered AVR on par with males with AR. To our knowledge, this is the first study to evaluate the effect of gender on survival in patients with severe AR with and without AVR.

## 6. Study Limitations

This is a retrospective observational study and, hence, is prone to limitations as treatment assignment is not random. Adjustments for covariate imbalances were made using Cox regression models.

## Figures and Tables

**Figure 1 medsci-11-00036-f001:**
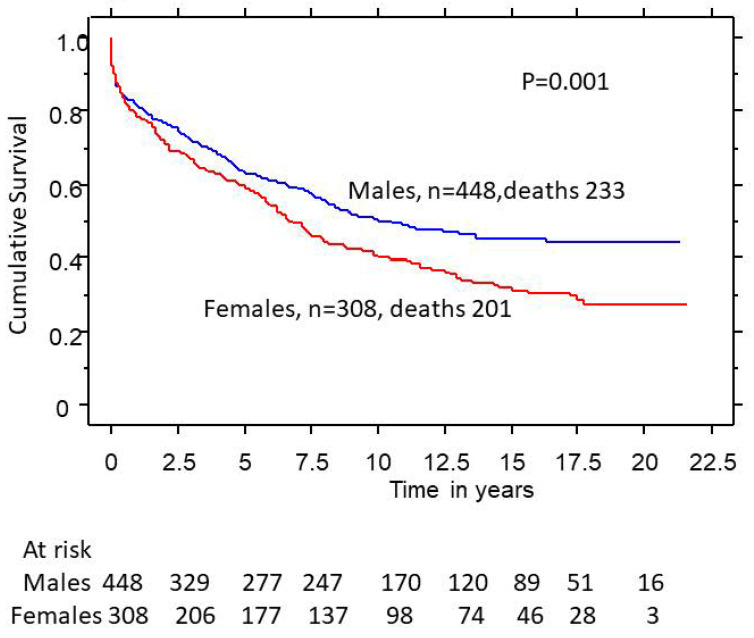
Survival in all severe aortic regurgitation as a function of gender.

**Figure 2 medsci-11-00036-f002:**
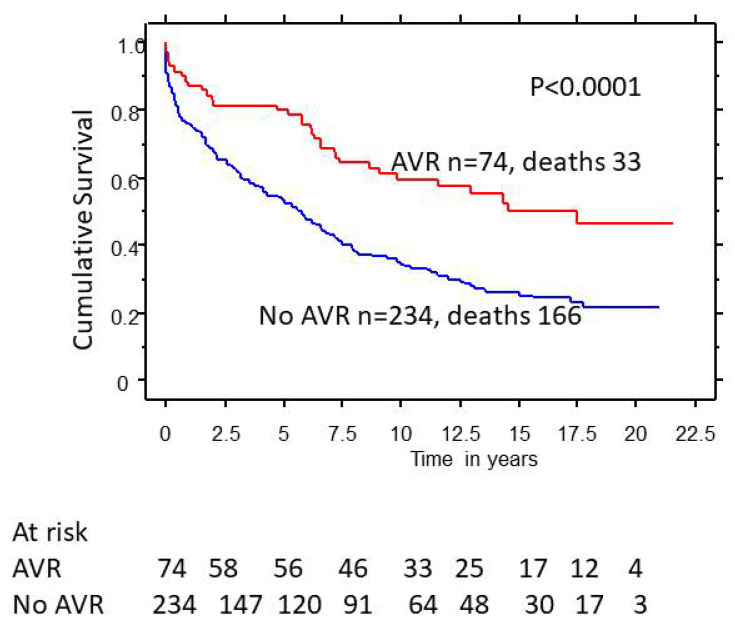
Effect of aortic valve replacement (AVR) on survival in female patients with severe aortic regurgitation (AR).

**Figure 3 medsci-11-00036-f003:**
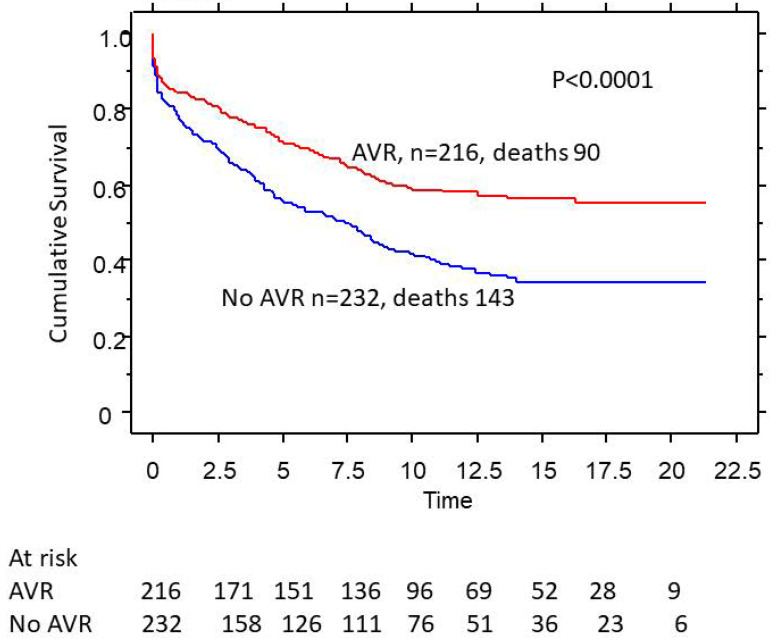
Effect of aortic valve replacement (AVR) on survival in male patients with severe aortic regurgitation (AR).

**Figure 4 medsci-11-00036-f004:**
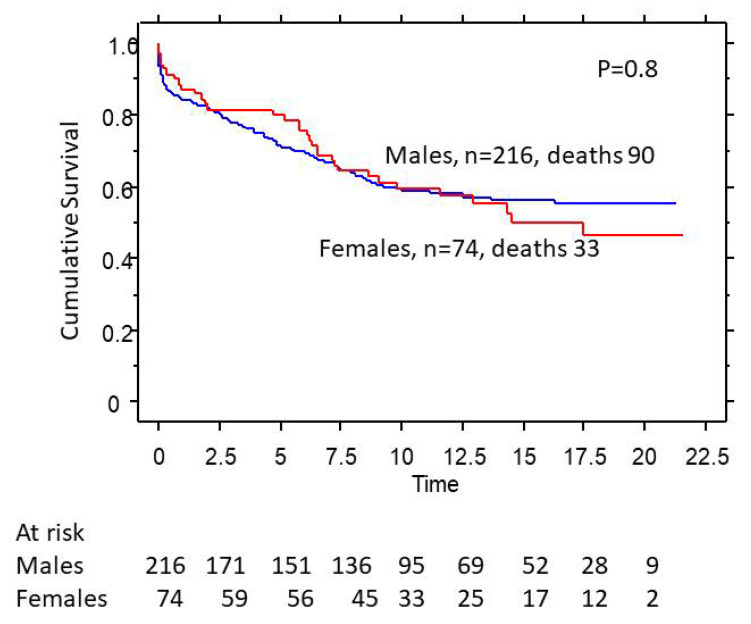
Effect of gender on survival in severe aortic regurgitation (AR) patients who had aortic valve replacement (AVR).

**Table 1 medsci-11-00036-t001:** Baseline characteristics as a function of gender.

Variables	Males (*n* = 448)	Females (*n* = 308)	*p* Value
Age in years	59 ± 17	64 ± 18	0.0002
LVEDD in cm	6.0 ± 1.0	5.2 ± 1.1	<0.0001
LVESD in cm	4.2 ± 1.2	3.6 ± 1.1	<0.0001
Septum in cm	1.3 + 0.2	1.2 + 0.3	0.0008
Posterior wall in cm	1.2 + 0.2	1.1 + 0.2	0.002
Ejection fraction%	52 ± 18	56 ± 17	0.003
Hypertension	62%	68%	0.18
Diabetes mellitus	11%	18%	0.006
Coronary artery disease	35%	32%	0.3
Heart failure	70%	69%	0.7
AV replacement	48%	24%	<0.0001
RVSP > 59	18%	15%	0.3
ACE or ARB	52%	48%	0.4
Calcium channel blockers	28%	31%	0.4
Beta Blockers	50%	57%	0.05
Greater than 2+ MR	40%	52%	0.0008

Abbreviations: ACE (angiotensin converting enzyme), ARB (angiotensin receptor blocker), AV (aortic valve), LVEDD (left ventricular end diastolic dimension), LVESD (left ventricular end systolic dimension), MR (mitral re-gurgitation), RVSP (right ventricular systolic press.

**Table 2 medsci-11-00036-t002:** Cox regression analysis of survival as a function of gender.

Variables	Exponential Coefficient	95% CI	*p* Value
Female gender	1.0	0.77–1.25	0.9
Age in years	1.038	1.02–1.04	<0.0001
LV EDD in mm	0.88	0.76–1.0	0.07
EF in %	0.99	0.98–0.99	0.009
Diabetes	1.82	1.35–2.37	<0.0001
AVR	0.74	0.58–1.04	0.02

Abbreviations: AVR (aortic valve replacement), EF (ejection fraction), LV EDD (left ventricular end diastolic dimension).

## Data Availability

Data sharing is not applicable to this article.

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
