# Peer review of "Gender Effects on Left Ventricular Responses and Survival in Patients with Severe Aortic Regurgitation: Results from a Cohort of 756 Patients with up to 22 Years of Follow-Up"

_medsci, 2023, doi:10.3390/medsci11020036_

Round 1
Reviewer 1 Report
The manuscript deals with clinically important topic- evaluation of the effect of gender on survival and remodeling responses in patients with severe aortic regurgitation (AR). Moreover, it is the first study to evaluate the effect of gender on survival in patients with severe AR with and without aortic valve replacement (AVR).
The introduction is consistent. The study design and result are presented shortly and clearly, including 2 tables and 4 figures. The study showed that females with severe AR were older, had differential remodeling characteristics such as smaller chamber sizes, had less thick ventricles, had better LVEF, and had lower rates of AVR compared to males. Female subjects had worse survival when compared to males on univariate analysis, which did not translate into statistical significance on multivariate analysis.
Discussion is rather short, but well written. Besides other data, the authors state, that the women with severe AR have differences in LV adaptive responses. It would be preferable if the authors expand their considerations, concerning the causes of different adaptive responses in male and female.
The limitations of the study are clearly presented. As it is a retrospective observational study, certain echocardiography data is under the question, i.e. ”the LV ejection fraction (EF) was assessed by planimetry or visually”-it means by “eyeballing”, it is not clear what percent of pts were assessed by “eyeballing” and what percent- by planimetry (Simpson’ s rule), it should be stated in the manuscript as well, having in mind that quantitative analysis is more preferable even in the observational studies.
The English level of the manuscript is high. The manuscript will be interesting to the wide auditorium of medical specialists, especially cardiologists, cardiac surgeons etc.
I recommend the manuscript for publication, owing to clinically relevant and up to date topic.
Author Response
We thank you and the reviewers for their comments and an opportunity to revise and resubmit our manuscript. We have our responses as noted below.
The manuscript deals with clinically important topic- evaluation of the effect of gender on survival and remodeling responses in patients with severe aortic regurgitation (AR). Moreover, it is the first study to evaluate the effect of gender on survival in patients with severe AR with and without aortic valve replacement (AVR).
The introduction is consistent. The study design and result are presented shortly and clearly, including 2 tables and 4 figures. The study showed that females with severe AR were older, had differential remodeling characteristics such as smaller chamber sizes, had less thick ventricles, had better LVEF, and had lower rates of AVR compared to males. Female subjects had worse survival when compared to males on univariate analysis, which did not translate into statistical significance on multivariate analysis.
We thank the reviewer for their comments.
Discussion is rather short, but well written. Besides other data, the authors state, that the women with severe AR have differences in LV adaptive responses. It would be preferable if the authors expand their considerations, concerning the causes of different adaptive responses in male and female.
Studies looking at the effect of gender on regurgitant valve lesions are sparse. Hence the discussion section is brief.
The limitations of the study are clearly presented. As it is a retrospective observational study, certain echocardiography data is under the question, i.e. ”the LV ejection fraction (EF) was assessed by planimetry or visually”-it means by “eyeballing”, it is not clear what percent of pts were assessed by “eyeballing” and what percent- by planimetry (Simpson’ s rule), it should be stated in the manuscript as well, having in mind that quantitative analysis is more preferable even in the observational studies.
We thank the reviewer for their comment. About 30% of the studies had images that were not suitable for quantifying ejection fraction by Simpson’s method. In these studies Visual estimation of ejection fraction was used.
The English level of the manuscript is high. The manuscript will be interesting to the wide auditorium of medical specialists, especially cardiologists, cardiac surgeons etc.
We thank the reviewer for their comment
I recommend the manuscript for publication, owing to clinically relevant and up to date topic.
We thank the reviewer for their comment.
Reviewer 2 Report
After examining the scientific study, the following considerations may be made. The scientific study is well structured in all its parts. In particular, the premises with which the authors introduced the analysis are clear. Equally clear are the objectives that led the authors to carry out this study and the section on materials and methods. Particular appreciation can also be expressed for the material on which the study was carried out. The data was collected methodically and without bias. The results were consistent and significant and allowed a discussion section full of food for thought. The authors then developed a discussion of the results achieved.
Author Response
We thank the reviewer for their comments